# Use of Biomarkers and Imaging for Early Detection of Pancreatic Cancer

**DOI:** 10.3390/cancers12071965

**Published:** 2020-07-19

**Authors:** Shingo Kato, Kazufumi Honda

**Affiliations:** 1Department of Clinical Cancer Genomics, Yokohama City University, Yokohama 236-0004, Japan; shin800m@yokohama-cu.ac.jp; 2Department of Biomarkers for Early Detection of Cancer, National Cancer Center Research Institute, Tokyo 104-0045, Japan; 3Department of Bioregulation, Graduate School of Medicine, Nippon Medical School, Tokyo 113-8602, Japan

**Keywords:** pancreatic cancer, early detection, blood biomarkers, fat-water MRI

## Abstract

Pancreatic cancer remains one of the deadliest cancers worldwide, and it is typically diagnosed late, with a poor prognosis. Early detection is the most important underlying factor for improving the prognosis of pancreatic cancer patients. One of the most effective strategies for detecting cancers at an early stage is screening of the general population. However, because of the low incidence of pancreatic cancer in the general population, the stratification of subjects who need to undergo further examinations by invasive and expensive modalities is important. Therefore, minimally invasive modalities involving biomarkers and imaging techniques that would facilitate the early detection of pancreatic cancer are highly needed. Multiple types of new blood biomarkers have recently been developed, including unique post-translational modifications of circulating proteins, circulating exosomes, microRNAs, and circulating tumor DNA. We previously reported that circulating apolipoprotein A2 undergoes unique processing in the bloodstream of patients with pancreatic cancer and its precancerous lesions. Additionally, we recently demonstrated a new method for measuring pancreatic proton density in the fat fraction using a fat–water magnetic resonance imaging technique that reflects pancreatic steatosis. In this review, we describe recent developments in potential biomarkers and imaging modalities for the early detection and risk stratification of pancreatic cancer, and we discuss current strategies for implementing screening programs for pancreatic cancer.

## 1. Strategy for the Early Detection of Pancreatic Cancer

### 1.1. Survival Rate and Incidence of Pancreatic Cancer

Pancreatic cancer remains one of the deadliest cancers worldwide. In the United States, the 5-year overall survival rate for all cancers diagnosed from 2009 through 2015 was 67%, whereas that for pancreatic cancer was only 9% [1]. In Japan, the 5-year survival rate for all cancers diagnosed from 2006 through 2008 was 62.1% overall, whereas that for pancreatic cancer was only 7.7% [2]. The results of an analysis of pancreatic cancer survival trends from 2007 through 2015 in Ontario, Canada, were recently reported. The authors of that study found no detectable improvement in survival in patients with pancreatic cancer involving regional or distant metastasis. However, for patients with localized pancreatic cancer, the 5-year survival rate increased during the study period from 22% to 61% [3]. Obviously, in addition to developing novel therapeutic strategies, the most important factor for increasing the survival rate of pancreatic cancer is early detection.

One of the most effective strategies for detecting cancers at an early stage is screening of the general population. However, as screening the general population is costly, this approach can be less cost-effective if the incidence of the cancer of interest is very low. Thus, determining the incidence of pancreatic cancer is important for developing an effective strategy for early detection. Data from the Global Cancer Observatory website (GLOBOCAN) [4] indicate that the incidence of pancreatic cancer varies across regions and populations. The estimated age-standardized incidence rate (ASR) for pancreatic cancer per 100,000 people in 2018 was 9.7 in Japan, 7.7 in the United States, and 6.4 in Canada. Although the ASR is useful for comparing cancer incidence rates between different countries, it can be misleading when the population includes a higher proportion of elderly persons compared to the global standard population, as is the case in Japan. For example, the although the actual incident rate of pancreatic cancer in Japan was 32.0 per 100,000 people in 2016 [5], the ASR of pancreatic cancer was calculated at 9.9 when normalized to the global standard population. In Japan, the government recommends general screening for several types of cancers, such as female breast cancer, lung cancer, prostate cancer, stomach cancer, and colon cancer. Compared to these types of cancers, the incidence of pancreatic cancer is relatively low. Therefore, when discussing or developing biomarkers for pancreatic cancer, this low incidence must be considered.

### 1.2. Strategy for Detecting Low-Incidence Cancers

In the general population, the lifetime risk of developing pancreatic cancer is estimated at approximately 1.6% based on United States data from 2015 to 2017 [1]. The International Cancer of the Pancreas Screening (CAPS) Consortium recommends targeted screening using magnetic resonance imaging (MRI) and endoscopic ultrasonography (EUS) for high-risk individuals having either >5% lifetime risk or a 5-fold increase in relative risk [6,7]. Indeed, the standardized incidence ratio for individuals in which there is familial pancreatic cancer in two first-degree relatives is reportedly 6.4-fold greater than that of the general population (95% confidential interval 1.8–16.4) [8]. However, it has been suggested that only 5–10% of pancreatic cancer patients have a familial basis [9], and most cases of pancreatic cancer are sporadic. Therefore, efficient screening strategies using a non-invasive modality to enrich high-risk individuals from the general population are urgently needed.

In August 2019, the US Preventive Services Task Force (USPSTF) updated their 2004 recommendations regarding screening for pancreatic cancer using imaging modalities. In the updated statement, the USPSTF did not recommend screening for pancreatic cancer in the general population using imaging methods [10]. In addition, they indicated that there are no current reports demonstrating validated biomarkers that enable the accurate detection of pancreatic cancer at an early stage [10]. However, importantly, this recommendation does not apply to persons with certain inherited genetic syndromes or a history of familial pancreatic cancer because they are at high risk for the disease. One of the reasons the USPSTF did not recommend general screening of asymptomatic persons is the low overall prevalence in the population [11,12,13]. Therefore, the targeted screening of subpopulations would be more worthwhile if it was possible to identify subpopulations in which there is a high prevalence of pancreatic cancer.

### 1.3. High Risk Individuals for Developing Pancreatic Cancer

The relative risk for developing pancreatic cancer in patients with known hereditary syndromes associated with an increased risk of developing pancreatic cancer is reportedly 132-fold greater than that of the general population [14]. Other than hereditary syndrome, family history itself is a risk factor for developing pancreatic cancer. Familial pancreatic cancer is defined in most studies as pancreatic cancer that develops in a family with two or more patients with pancreatic cancer among their parents or siblings (first-degree relatives). In familial pancreatic cancer, the first-degree relatives of patients reportedly have a 9-fold increased risk of pancreatic cancer compared to the general population [8].

Investigations of genomic variation or mutations (germ-line mutations) have recently become more important and common in the clinical setting. The development of pancreatic cancer reportedly involves germ-line mutations in multiple genes, such as *BRCA1*/2, *PALB2*, *ATM*, *CDKN2A*, *APC*, *MLH1*, *MSH2*, *MSH6*, *PMS2*, *PRSS1*, and *STK11* [15,16]. The National Comprehensive Cancer Network guideline recommends germ-line testing for any patient with confirmed pancreatic cancer using comprehensive gene panels for hereditary cancer syndromes [17]. Genetic testing has also become more important for pancreatic cancer patients, as those having *BRCA1/2*-mutant tumors are predicted to exhibit increased therapeutic sensitivity to platinum-containing therapeutic agents and inhibitors of poly-(ADP-ribose)-polymerase (PARP) [18].

Multiple other risk factors related to lifestyle and underlying disease have also been reported. For example, smoking [19,20], heavy alcohol consumption [21,22], and obesity [23,24] are reportedly risk factors for pancreatic cancer. In terms of underlying disease, longstanding pre-existing chronic pancreatitis and new-onset diabetes mellitus are important. Over the past several decades, research has revealed that chronic pancreatitis is a well-defined risk factor of pancreatic cancer and that development of the disease requires 30-40 years of inflammation [25,26,27]. An association between both types I and II diabetes mellitus and the development of pancreatic cancer has been reported in multiple studies [28,29,30], and new-onset diabetes mellitus is expected to be used as a marker for pancreatic cancer. These two underlying diseases related to the development of pancreatic cancer represent detectable clinical markers for the enrichment of high-risk populations.

## 2. Precursor Lesions of Pancreatic Cancer

The early detection and treatment of cancer has been made possible by understanding the precancerous lesions associated with each cancer type [31,32,33]. With respect to pancreatic cancer, two important precursor lesions have been identified: pancreatic intraepithelial neoplasia (PanIN) and intraductal papillary mucinous neoplasm (IPMN) [34,35].

### 2.1. Pancreatic Intraepithelial Neoplasia (PanIN)

PanIN is a non-invasive macroscopic lesion that occurs in the small pancreatic epithelium. Three types of PanIN were originally described (PanIN-1 to PanIN-3) in 2001 based on the degree of cellular and structural abnormalities [34]. In 2015, this classification was revised to a two-tiered system consisting of low-grade PanIN (originally, PanIN-1/2) and high-grade PanIN (originally, PanIN-3) [36]. As PanIN is a microscopic change, these lesions are difficult to detect using current non-invasive imaging modalities.

Although the natural history of PanINs cannot be monitored in living individuals, analyses using autopsy and surgically resected samples have provided several characteristics of the lesions. The number and severity of PanINs are reportedly associated with aging [37,38] and pancreatic fibrosis [39,40]. Additionally, obesity and pancreatic fat accumulation are reportedly risk factors for PanIN [41]. In recent autopsy analyses of 173 cases exhibiting no evidence of pancreatic cancer, PanIN-3 was found in 4% of examined cases, whereas PanIN-1 and PanIN-2 were present in 77% and 28%, respectively [42]. As the authors reported, the average age of patients (80.5 years) included in their study was much older than that of patients with pancreatic ductal adenocarcinoma (PDAC), and patients with PDAC and/or IPMN were excluded from the study cohort. Thus, the biology of PanIN-3 lesions in their study may be different from that observed in association with PDAC. Although the exact proportion of PanIN patients who eventually develop invasive cancer remains unknown [43,44], the development of a novel method to detect these PanINs in living individuals would be of great value in the early diagnosis of pancreatic cancer.

### 2.2. Intraductal Papillary Mucinous Neoplasms (IPMNs)

IPMN is an epithelial pancreatic neoplasm that produces mucin and forms cystic lesions. IPMN represents a broad group of pathologies. The grading system for IPMNs was updated in 2015 [36], such that IPMNs are now classified as either low-grade or high-grade. The terminology “carcinoma in situ” is accepted for indicating high-grade IPMN [45].

IPMNs can arise from both the main and branch pancreatic ducts. Due to the increasing use of cross-sectional imaging in recent years, IPMNs are frequently detected incidentally [13]. As main duct and combined-type IPMN exhibit a high rate (31% to 45.5%) of progression to invasive malignancies, surgical resection is commonly indicated [12,43,46]. In contrast, branch duct IPMNs exhibit lower rates (2.9% to 15%) of progression to invasive malignancy [46,47]. For high-risk cases of branch duct IPMN, imaging features indicative of high risk and recommendations regarding surveillance have been reported [48].

Although both PanIN and IPMN are non-invasive epithelial neoplasms arising in the pancreatic duct, only IPMNs exhibit a morphologic lesion that can be detected by non-invasive imaging. Therefore, the presence of IPMNs is one of the few clues enabling the early detection of pancreatic cancer.

## 3. Potential Blood Biomarkers for Early Detection and Risk Stratification of Pancreatic Cancer

As most patients with early stage pancreatic cancer are asymptomatic, establishing a general screening system is the only way to enable early detection. To this end, multiple blood biomarkers are being developed. However, to date, no biomarkers have been approved for the early diagnosis of pancreatic cancer. Therefore, blood tests that enable the detection of resectable pancreatic cancer and/or its precursor in individuals within the asymptomatic population with or without hereditary factors are urgently needed.

### 3.1. Carbohydrate Antigen 19-9 (CA19-9)

The glycan carbohydrate antigen 19-9 (CA19-9) is one of the most important and widely used blood biomarkers in the diagnosis of pancreatic cancer. CA19-9 was first described in 1979 as a tumor antigen recognized by the monoclonal antibody NS19-9 [49]. Elevated levels of CA19-9 are an important finding for the diagnosis of pancreatic cancer, and changes in CA19-9 levels are useful for monitoring treatment response. However, CA19-9 is not considered an appropriate marker for the mass screening of asymptomatic, early stage pancreatic cancer [50]. Several studies have evaluated mass screening for pancreatic cancer among asymptomatic individuals using CA19-9. Satake et al. assessed serum CA19-9 levels in 10,162 asymptomatic individuals in Japan and detected 4 pancreatic cancers (1 resectable) among 18 asymptomatic patients (0.2%) with an elevated CA19-9 serum level [51]. Kim et al. examined serum CA19-9 levels in 70,940 asymptomatic individuals in Korea and identified 4 patients with pancreatic cancer among 1063 patients with elevated CA19-9 serum levels [52]. Similarly, Chang et al. screened 5343 asymptomatic individuals in Taiwan and identified 2 patients with pancreatic cancer among 385 patients with elevated CA19-9 serum levels [53]. These results are summarized in Table 1.

Several limitations preclude the use of CA19-9 as an early detection marker for pancreatic cancer. For example, CA19-9 levels can be elevated in several benign diseases involving not only the pancreas and biliary tract [54] but also other organs [55]. In addition, CA19-9 levels can be elevated in multiple types of advanced gastrointestinal adenocarcinoma [56]. More importantly, elevated levels of CA19-9 may have only limited sensitivity for detecting pancreatic cancer lesions at the small, curable stage [57]. Furthermore, CA19-9 is not expressed in all individuals, as 5–10% of the population cannot produce CA19-9 due to the lack of enzymes essential for its synthesis [58]. Based on these limitations, CA19-9 is not currently recommend as a marker for the early detection of pancreatic cancer, according to the American Society for Clinical Oncology (ASCO) Provisional Clinical Opinion [59].

### 3.2. Exosomes and MicroRNA

Exosomes are a type of extracellular vesicle (EV) that contains cytoplasmic components, including nucleic acids and soluble proteins. Several kinds of EVs have been described, and they are categorized based on size and biogenesis mechanism [60]. Exosomes were first described in 1987 as having a diameter of less than 100 nm [61,62]. Exosomes can be isolated from multiple body fluids, including plasma [63], serum [64], urine [65], saliva [66], and breast milk [67]. As exosomes released by cancer cells may contain tumor-specific components, they are important candidate biomarkers for early detection. However, the use of exosomes as screening biomarkers is limited by the time-consuming and labor-intensive procedures needed to prepare samples. For example, the standard method for isolating exosomes includes ultracentrifugation at 15,000× *g* overnight at 4 °C, followed by several additional steps [68]. To overcome this limitation, multiple less time-consuming purification methods have been developed [69]. Although the consistency of each new method in comparison with the conventional method should be verified, these alternative methods are expected to enable the use of circulating exosomes as a screening marker in the future.

MicroRNAs (miRNAs) are functional nucleic acids encoded in the genome that undergo a multi-step process that ultimately forms a small RNA of 20 to 25 bases in length [70]. miRNAs belong to a class of non-coding RNAs that regulate gene expression via mRNA degradation or the inhibition of translation [71]. As nucleases are present at high concentrations in serum and plasma, the half-life of naked miRNAs is only several minutes. However, stable miRNAs can be detected in serum and plasma, as circulating miRNAs can form various complexes with protective proteins or be incorporated into EVs [72,73,74,75].

Analyses of various tissue samples revealed that several miRNAs, such as miR-21, miR-155, and mi-R 196, are upregulated in pancreatic cancer lesions compared to normal or re-cancerous lesions [76,77,78,79]. Other studies demonstrated that the miRNAs miR-17-5p [80], miR-21 [80], miR-10b [81], miR-550 [82], and miR-451a [83] are upregulated in the blood of pancreatic cancer patients. However, these studies primarily used blood samples from patients with unresectable pancreatic cancer.

Xu et al. examined whether exosome miRNA could be used as an indicator of localized (stage I and stage IIA) pancreatic cancer [84]. They collected exosomes from the plasma samples of patients with localized pancreatic cancer (stage I-IIA, *n* = 15) and healthy subjects (*n* = 15) and analyzed plasma exosome miRNA expression using qRT-PCR. They found that plasma exosome miR-196a and miR-1246 levels were significantly elevated in pancreatic cancer patients as compared to healthy subjects. The area under the curve (AUC) was 0.81 (95% confidence interval [CI] 0.64, 0.97; *p* < 0.001) for miR-196a and 0.73 (95% CI 0.54, 0.92; *p* = 0.019) for miR-1246. Although the fold changes in relative expression levels relative to the control subjects were small (<2 in miR-196a and <3 in miR-1246), this study demonstrated the possibility of using miRNAs as early detection markers for pancreatic cancer.

### 3.3. Circulating Tumor DNA

Cell-free DNA (cfDNA) is composed of non-encapsulated DNA fragments circulating in the bloodstream. These fragments are released into the bloodstream from dying cells and usually removed by macrophages. The average length of a cfDNA is approximately 170 bases, with a half-life of approximately 2 h [85].

The strategy for using cfDNA as an early cancer detection marker is the detection of cancer cell-derived cfDNA (circulating tumor DNA; ctDNA) among the total cfDNA. However, this approach has several technical difficulties. Although it was reported that cancer patients have 4 to 40 times higher levels of cfDNA than steady-state healthy individuals due to the overproduction of cancerous cells [86], increased levels of total cfDNA occur in other conditions as well, such as inflammation, trauma, and after exhausting exercise [87,88,89,90]. As the amount of background cfDNA from normal cells increases, the signal-to-noise ratio decreases, making the detection of ctDNA difficult. Increased levels of background cfDNA from normal cells can be caused by not only the specific condition of the patient but also by the sample preparation procedure. After the blood sample is obtained, cfDNA increases over time due to the death of normal cells. Thus, if an analysis of cfDNA is planned, blood samples should be separated from live cells immediately after collection. Collectively, these data suggest that if ctDNA is used for general screening, standard procedures for sample collection and preparation should be strictly followed.

In using ctDNA as a detection biomarker, a target gene for analysis should be identified. Ideally, it is desirable to target a gene that is mutated in a majority of pancreatic cancer cases. The term “pancreatic cancer” indicates a malignant tumor that occurs in the pancreas. The majority (85–90%) of these cancers are PDAC, and other subtypes are relatively rare [91,92]. It is well-established that the majority of PDAC cases involve mutation in the *KRAS* gene [93]. Multiple independent sequencing analyses involving a large number of PDAC patients showed similar results, with approximately 90% of PDACs carrying at least one *KRAS* mutation [94,95]. Moreover, *KRAS* mutations are the earliest genetic alteration driving pancreatic neoplasia [96,97,98]. Therefore, *KRAS* is usually selected as the target gene in screening for pancreatic cancer ctDNA.

Le Calvez-Kelm et al. investigated the utility of detecting *KRAS* mutations in plasma cfDNA as a marker for pancreatic cancer in a large case-control series [99]. They analyzed plasma samples of 437 pancreatic cancer patients, 141 subjects with chronic pancreatitis, and 394 healthy controls. Although an amplicon-based deep-sequencing technology was used to improve the sensitivity of detecting low-abundance mutations, the detection rate for *KRAS* mutations in cfDNA was 10.3%, 17.5%, and 33.3% for samples from patients with local, regional, and systemic stages, respectively. Additionally, cfDNA *KRAS* mutations were also detected in 3.7% of samples from healthy controls and 4.3% of samples from chronic pancreatitis patients. Cohen et al. also assessed *KRAS* mutations in cfDNA in 221 patients with resectable PDAC and 182 control patients without known cancer [100]. To improve the sensitivity, they used a PCR-based parallel sequencing approach called the “Safe-Sequencing System” (i.e., Safe-SeqS) [101]. However, the detection rate for *KRAS* mutations in cfDNA was only 30% in the PDAC patients [100]. In addition, *KRAS* mutations were detected in one sample from the control group (0.6%) [100].

At this time, even with advanced sequencing technology, the identification of *KRAS* mutations in cfDNA exhibits limited sensitivity as an early detection marker for pancreatic cancer. However, recently, analyzing a patient’s cancer genome has become a common clinical test [102,103]. Accumulating data from these analyses could facilitate the identification of new pancreatic cancer biomarker candidates other than *KRAS*.

### 3.4. Multi-Analyte Blood Test

As discussed above, analyzing mutations in one gene using cfDNA reportedly has insufficient sensitivity to detect early stage pancreatic cancer. This sensitivity could be improved by increasing the number of analyzed genes; however, such analyses are costly. To implement the analysis of cfDNA as a marker for detecting early stage cancers, Choen et al. developed a blood test algorithm known as CancerSEEK, which is capable of detecting eight common types of cancer, including pancreatic cancer [104]. From analyses of publicly available cancer patient cfDNA sequencing data, they found a fractional power law relationship between the number of amplicons required and the sensitivity of cancer detection. Importantly, the cancer detection rate plateaued at approximately 60 amplicons. Based on this finding, Choen et al. designed a 61-amplicon panel as the minimum number of short amplicons required to detect at least one cancer-related gene mutation. Combining sequence data for these 61 sites in cfDNA with the levels of eight blood proteins, they developed the CancerSEEK algorithm. The CancerSEEK test was applied to 1005 patients with non-metastatic, clinically detected cancers of the ovary, liver, stomach, pancreas, esophagus, colorectum, lung, or breast. Of the 1005 patients included in the analysis, 93 had pancreatic cancer. The CancerSEEK test was positive in 67 of the 93 patients with pancreatic cancer; 5 of 6 in stage I, 2 of 10 in stage II, and 5 of 10 in stage III were positive.

Another group reported the multiparametric analysis of blood samples to develop a novel biomarker signature of early stage PDAC [105]. Using an antibody microarray platform, they created a biomarker signature that could distinguish blood samples obtained from patients with stage I and II PDAC from those of samples obtained from controls. The created signature consisted of 29 biomarkers and could discriminate patients with stage I and II PDAC from controls with an AUC value of 0.96. As in these studies, combining multiple biomarkers using bioinformatics technology is expected to expand in the future.

### 3.5. Apolipoprotein A2 Isoforms

Circulating serum/plasma proteins and peptides are known to undergo unique post-translational modifications specific to the molecular pathology of each disease. We recently identified a unique post-translational modification of circulating apolipoproein A2 (apoA2) homodimers that can be used to efficiently detect patients with PDAC who can undergo resectable surgery and also risk-stratify PDAC patients. We reported that apoA2-isoforms exhibit a unique processing pattern in pancreatic diseases, including PDAC [106,107,108]. ApoA2 is a major component of high-density lipoproteins (HDLs), and five isoforms of ApoA2 have been identified to date [106,107,108,109].

Circulating apoA2 forms a homodimer, and it consists of 5 isoforms with different C-terminal amino acid sequences: -ATQ/ATQ, -ATQ/AT, -A2-AT/AT, -A2-AT/A, and -A2-A/A. Based on molecular weight, we named the isoforms apoA2-ATQ/ATQ, apoA2-ATQ/AT, apoA2-AT/AT, apoA2-AT/A, and apoA2-A/A. In the bloodstream of healthy subjects, 3 basic isoforms are typically detected in well-balanced proportion: apoA2-ATQ/ATQ, apoA2-ATQ/AT, and apoA2-AT/AT. In contrast, blood samples from patients with PDAC typically contain two aberrantly processed apoA2 isoforms of higher or lower molecular weight, with unique apoA2-isoform processing patterns observed. We defined these as a hypo-processing pattern, in which a heavy isoform (apoA2-ATQ/ATQ) is predominantly recognized, and a hyper-processing pattern, in which light isoforms (apoA2-AT/AT, apo-AT/A, and/or apoA2-A/A) are predominantly recognized [107,108]. In both processing patterns, the level of circulating apoA2-ATQ/AT, the intermediate-weight isoform, is significantly lower in the bloodstream of patients with PDAC in comparison with healthy controls (Figure 1). Although the detailed mechanisms underlying these aberrant processing patterns remain unknown, they may be related to pancreatic exocrine disorder [110,111]. The pancreas synthesizes carboxypeptidase A, which is an enzyme that cleaves peptide bonds at the C-terminus of proteins or peptides. The abnormal expression or release of carboxypeptidase A may lead to a reduction in plasma levels of apoA2-ATQ/AT, which is the major intermediate apoA2 isoform, in comparison with the normal state.

To simplify the measurement of blood concentrations of the intermediate apoA2-ATQ/AT isoform, we developed an enzyme-linked immunosorbent assay (ELISA) in 2015 [108]. A study to validate this novel assay for pancreatic cancer detection was conducted jointly in Japan in conjunction with the US National Cancer Center Early Detection Research Network (NCI EDRN). In that study, we found that combining an analysis of this apoA2 isoform assay with the determination of CA19-9 significantly improved the diagnostic accuracy compared with an analysis of CA19-9 alone [108]. The AUC values for CA19-9 and apoA2-ATQ/AT isoform as single biomarkers to distinguish patients with early stage pancreatic cancer (stage I/II) were 0.783 (95% CI 0.64, 0.95, 0.699–0.855), and 0.809 (95% CI, 0.748–0.867), respectively. Additionally, a prospective evaluation was conducted to measure CA19-9 and apoA2-ATQ/AT levels in 156 patients with pancreatic cancer and 217 matched controls within the European Prospective Investigation into Cancer and Nutrition (EPIC) study. The study included plasma samples collected up to 60 months prior to diagnosis. We found that the combined analysis of CA19-9 and apoA2-ATQ/AT could lead to the detection of pancreatic cancer up to 18 months prior to diagnosis under usual care compared to an analysis of CA19-9 alone [112].

The two aberrant processing patterns of apoA2 isoforms are also observed in individuals at high risk for developing PDAC, and analyses of these isoforms enables an efficient detection of IPMN, which is considered a precancerous lesion, as well as other pancreatic conditions (e.g., chronic pancreatitis) that predispose patients to pancreatic cancer [108,109]. These findings suggest that apoA2 isoforms are clinically useful as potential biomarkers for screening the general population for individuals at higher risk for PDAC prior to the use of imaging modalities. This approach could increase the positive predictive value of secondary screening approaches using imaging modalities such as EUS and magnetic resonance cholangiopancreatography (MRCP) for detecting pancreatic cancers, including PDAC. In addition, combining biomarker analyses with imaging could be an efficient screening strategy for PDAC. To test this hypothesis, we initiated a prospective clinical pancreatic screening study using blood tests for apoA2 isofoms in Japan.

## 4. Possible Non-Invasive Imaging Modalities for Early Detection of Pancreatic Cancer

Imaging examinations are currently necessary for the diagnosis of pancreatic cancer. The imaging modalities commonly used to diagnose pancreatic cancer include transabdominal ultrasound, CT, MRI, and endoscopic techniques such as EUS and endoscopic retrograde cholangiopancreatography (ERCP). However, the endoscopic techniques are not suitable for large-scale screening because the examinations are relatively invasive and time consuming. As stated in Section 1, the enrichment of high-risk individuals from the general population using non-invasive biomarkers is the first step in effective pancreatic cancer screening. Additionally, in order to develop efficient screening methods for pancreatic cancer, optimizing combinations of biomarkers and imaging modalities should be considered. The initial recommendation for high-risk individuals involves targeted imaging screening using a minimally invasive modality. This would be followed by more invasive examinations such as EUS via fine-needle aspiration (EUS-FNA) or ERCP to obtain tissue or cell samples for pathologic diagnosis.

### 4.1. Transabdominal Ultrasound

Transabdominal ultrasound (US) is an imaging modality that allows non-invasive observation of abdominal organs. The reported sensitivity and specificity of transabdominal US for the detection of pancreatic cancer range from 75% to 89% and 90% to 99%, respectively [113,114]. Recently, Kanno et al. reported the results of a multi-center retrospective study of 200 cases of early stage pancreatic cancer, including 51 cases of stage 0 and 149 cases of stage I disease [115]. Solid tumors were detected by transabdominal US in 68/101 (67.3%) stage I cases, in 96/146 (65.8%) cases by CT, 73/127 (57.5%) cases by MRI, and 122/132 (92.4%) cases by EUS. The limitation of transabdominal US for the general screening for pancreatic cancer is that the diagnostic performance of the modality greatly depends on the operator’s experience and the patient’s condition, presence of comorbidities such as obesity, and the presence and amount of interfering bowel gas.

### 4.2. CT

CT is a less invasive imaging modality, and multiple studies have reported that CT is useful for mass screening to detect early stage lung cancer [116,117]. With respect to mass screening for pancreatic cancer using CT, one of the advantages of CT is the short exam time, which enables the acquisition of many images. A recent meta-analysis of 15 studies including 815 PDAC patients reported that sensitivities for the CT-based detection of PDAC are 90% (95% CI, 87–93%), whereas specificities and accuracy were 87% (95% CI, 79–93%) and 89% (95% CI, 85–93%), respectively [118]. Obviously, smaller tumors are harder to detect and diagnose; therefore, the detection of small PDAC tumors requires an expert review of the imaging data [119]. In the case of CT, the use of contrast-enhancing agents is required to maximize sensitivity and diagnostic accuracy. As the possibility of adverse effects caused by these agents must be considered, CT exams for this purpose become slightly more invasive. Compared to endoscopic examinations and contrast-enhanced CT, MRI is less or non-invasive, except for claustrophobic patients and individuals with metallic implants that are not compatible with MRI.

### 4.3. MRI

MRI is often used for the diagnosis of pancreatic cancer [120]. One of the advantages of MRI is that the exam can incorporate multiple imaging methods through the use of contrast-enhancing agents. For example, diffusion-weighted MRI (DW-MRI), a contrast-enhancing agent-free method used to quantify tissue microstructures, is reportedly helpful for the diagnosis of pancreatic cancer [121,122,123,124]. MRCP, another MRI method, enables the visualization of pancreatic duct morphology. MRCP can aid in the detection of indirect signs of pancreatic cancer, such as dilation of the pancreatic ducts.

The reported sensitivity of MRI for the detection of PDAC is 93% (95% CI, 88–96%), as calculated from pooled data from 11 studies that included 349 PDAC patients, whereas the specificity and accuracy were 89% (95% CI, 82–94%) and 90% (95% CI, 86–94%), respectively [118]. However, compared to CT, MRI has greatly benefited from recent technological advances [125] that are not well reflected in older studies. However, these advanced techniques may not be suitable for mass screening, as some of them require considerable time to obtain images.

### 4.4. Three-Dimensional Analysis of Pancreatic Fat Using Fat–Water MRI

The primary goals of imaging for the detection of early pancreatic cancer are the detection of solid masses, indirect signs of cancer, and precursor lesions [120]. In addition, it is important to identify individuals with risk factors for pancreatic cancer and introduce them to further testing. As pancreatic steatosis is reportedly a risk factor for pancreatic cancer [126,127], we recently developed a new method for measuring pancreatic fat using MRI [128]. Determination of the proton density fat fraction (PDFF) using MRI enables calculation of the fatty rate of regions of interest (ROIs) set in the MRI scan [129]. Several groups have applied this MRI-based PDFF measurement method to evaluate the fatty rate of the pancreas [130,131,132,133,134]; however, in most of the studies, pancreatic PDFF was calculated from multiple small ROIs set in the pancreas. As fat accumulates in a patchy manner in the pancreas [135], PDFF values can differ widely depending on the locations of the ROIs [128] (Figure 2A). Therefore, pancreatic PDFF values calculated from small ROIs may not represent the true degree of pancreatic steatosis.

Based on the above background, we developed a novel method for determining the pancreatic PDFF that calculates average fat fraction values based on all the voxels included in the three-dimensional pancreas (Figure 2B). Our method facilitates the evaluation of pancreatic steatosis with greater repeatability and reproducibility. As stated in the “Apolipoprotein A2 isoforms” section, pancreatic exocrine disorder may be related to the levels of specific serum apoA2 isoforms [110,111]. Therefore, macroscopic changes in the pancreatic parenchyma, such as atrophy of acinar cells, could serve as imaging biomarkers in the future.

## 5. Future Prospects

The advantages and disadvantages of each biomarker and early detection method introduced in this review are summarized in Table 2. To decrease the mortality rate of pancreatic cancer, more efficient clinical screening methods are needed. The development of biomarkers and imaging modalities that enable the early detection of pancreatic cancer has been put on the fast track. However, since the incidence of pancreatic cancer is very low, it would be quite difficult to develop a screening method that has enough sensitivity and specificity for the early detection of pancreatic cancer by a single modality. Therefore, as in multi-analyte blood tests, combining multiple existing analyses is expected to solve this problem.

Not only the combination of multiple blood tests but also a combination of blood tests and radiologic images has been studied. For example, an approach that combines blood testing of miRNA or cfDNA with radiographic features has been reported [136]. Such so-called *-omics* approaches to radiology are commonly called radiomics [137], and when it includes genetic parameters, it is called radiogenomics. Although further validations are needed, such radiomics analyses are expected to emerge in the future to facilitate the early detection of pancreatic cancer.

However, even if these combination analyses demonstrate high performance, their use in screening the general population may not be feasible when considering their cost-effectiveness. Thus, we need to switch our strategy to the development of a sequential combination of tests. The pool of individuals at high risk within the general population can be enriched using non-invasive biomarkers such as liquid biopsy, with subsequent screening of high-risk individuals using imaging techniques. Imaging modalities should be selected based on comparisons of cost and invasiveness versus effectiveness. If the performance of a blood test used as the first screening technique is quite good, we could proceed directly to more invasive examinations. However, if the performance is poor, less invasive cross-sectional imaging modalities, such as CT or MRI, may be better choices as the first imaging modality. Detailed high-resolution imaging surveys of subjects with high-risk diseases such as IPMN and pancreatitis could enable the detection of pancreatic cancer at a sufficiently early stage to permit curative resection. There is an urgent need to develop screening techniques combining biomarkers and imaging modalities that are non-invasive or minimally invasive to enable the early detection of pancreatic cancer.

## Figures and Tables

**Figure 1 cancers-12-01965-f001:**
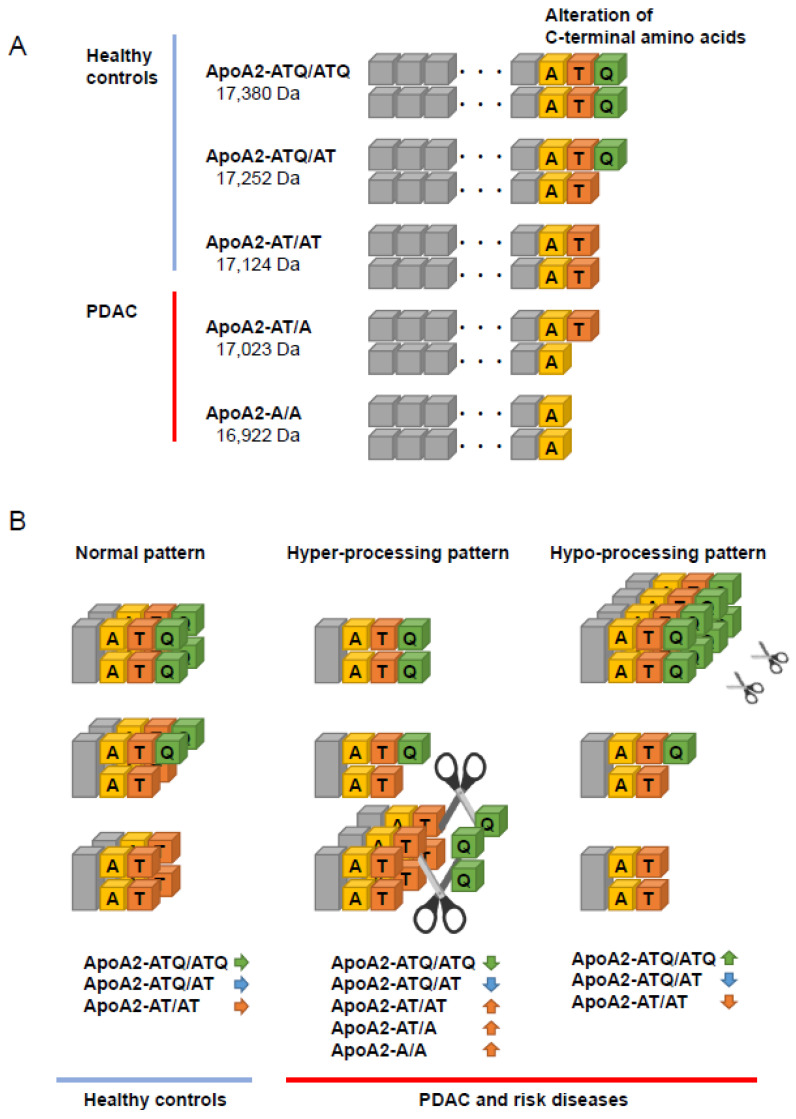
(**A**) Alteration of C-terminal amino acids of the apolipoproein A2 (apoA2) isoforms. The theoretical molecular weights of the five isoforms of apoA2 are shown at left. (**B**) The normal apoA2 isoforms were primarily distributed in healthy controls. However, the hypo-processing pattern, in which apoA2-ATQ/ATQ was predominantly expressed, and the hyper-processing pattern, in which apoA2-AT/AT was predominantly expressed, were detected in patients with pancreatic ductal adenocarcinoma (PDAC) or its risk-conferring diseases. In both the hyper- and hypo-processing patterns, because apoA2-ATQ/ATQ or apoA2-AT/AT was increased, apoA2-ATQ/AT consequently decreased in PDAC [109].

**Figure 2 cancers-12-01965-f002:**
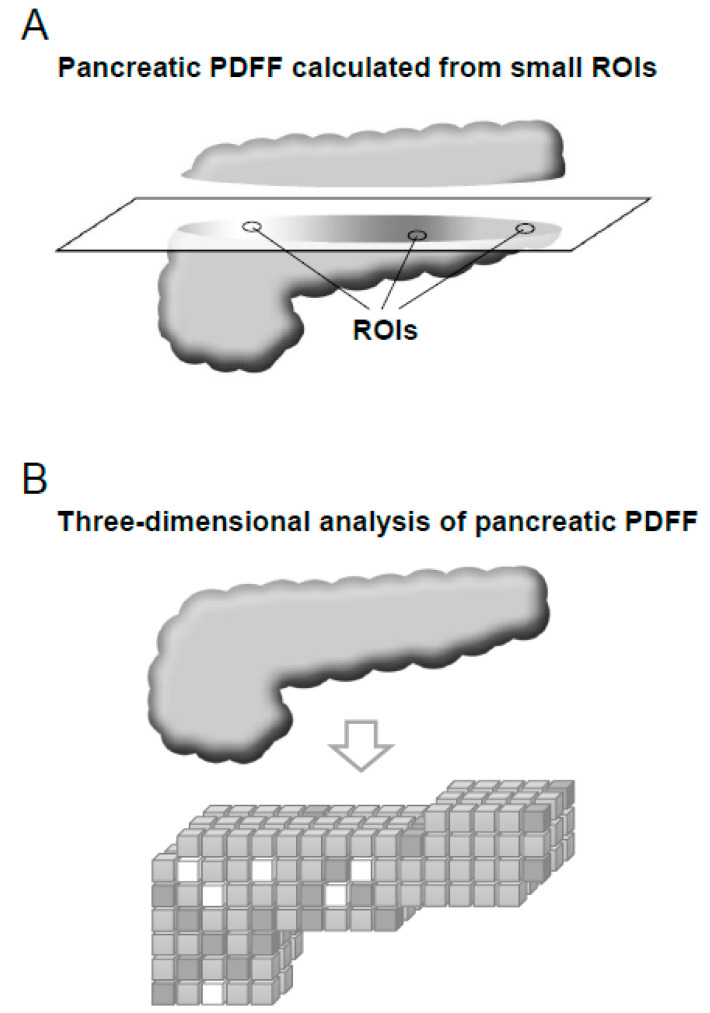
(**A**) Pancreatic proton density fat fraction (PDFF) calculated from multiple small pancreatic regions of interest (ROIs). The PDFF values differed widely depending on the ROI location. (**B**) Three-dimensional analysis of pancreatic PDFF. The average PDFF was calculated from all voxels included in the three-dimensional pancreas [128].

**Table 1 cancers-12-01965-t001:** General screening of asymptomatic individuals based on serum levels of CA19-9.

Reference	Publication Year	Country/Area	Total Number Examined	Number of Individuals with Elevated CA19-9	Detected PDAC Patients
Satake [51]	1994	Japan	10,162	18	4
Kim [52]	2004	Korea	70,940	1063	4
Chang [53]	2006	Taiwan	5343	385	2

**Table 2 cancers-12-01965-t002:** Advantages and disadvantages of each biomarker and early detection method.

Modalities	Cost	Invasiveness	Advantages	Disadvantages
Biomarker				
CA19-9	Low	Low	Low cost, easy to handle. Validated by multiple studies. Useful for monitoring treatment response.	Limited sensitivity [57]. 5–10% of the population cannot produce CA19-9 [58]. Currently not recommend as an early detection marker for pancreatic cancer by ASCO recommendations [59]. Unable to detect high-risk individuals.
Exosomes and microRNA	High	Low	Samples other than serum proteins, such as non-coding RNA, can be analyzed.	Sample preparation requires considerable effort [68]. Limited sensitivity in currently reported studies [84]. Insufficient validation as an early detection marker.
Circulating tumor DNA	Middle to High	Low	Multiple settings can be created; PCR base or next-generation sequencing (NGS) base.	Limited sensitivity in currently reported studies [99]. NGS base analysis is costly. Insufficient validation as an early detection marker.
Multi-analyte blood test	High	Low	Increased sensitivity and specificity compared to single-parameter analyses.	High cost. Multiple parameters are required to obtain one result. Insufficient validation as an early detection marker.
Apolipoprotein A2 isoforms	Low	Low	Low cost, easy to handle. Validated by multiple studies as an early detection marker [108,112]. Able to detect some high-risk individuals [108,109]. May be able to use a marker for pancreatic exocrine disorder [110,111].	Not validated as a marker for monitoring treatment response. Differentiation between high-risk individuals and patients with cancer may be difficult by itself.
Early detection methods				
Transabdominal ultrasound	Low	Low	Low cost. Non-invasive.	Unable to examine the whole pancreas. Observation quality depends on the operator’s skill. Sensitivity varies depending on the patient’s condition.
CT	Middle	Low-middle	Low cost. Minimally invasive. Results not affected by the operator’s experience. Examination is completed in a short time.	Low sensitivity for small (<10 mm) tumors [119]. Radiation exposure. May be invasive if contrast-enhancing agents are used.
MRI	Middle	Low	Low cost. Non-invasive. Results not affected by the operator’s experience. Helpful for the detection of indirect signs of PDAC. Additional information such as DW-MRI or PDFF can be obtained [121,122,123,124,128].	Individuals with metallic implants or with claustrophobia cannot be tested. Examination time is longer than CT.

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
