# Peer review of "Use of Biomarkers and Imaging for Early Detection of Pancreatic Cancer"

_cancers, 2020, doi:10.3390/cancers12071965_

Round 1

Reviewer 1 Report

The present manuscript by Shingo Kato and Kazufumi Honda is an updated review of potential biomarkers and imaging modalities for screening and early detection of pancreatic cancer. The manuscript is a well written and comprehensive review of the topic. Analysed aspects include CA19-9, exosomes and microRNA, circulating tumor DNA, circulating tumor cells, multi-analyte blood test, apolipoprotein A2 isoforms, and MRI techniques.

The most important papers are cited and discussed. Somewhat missing is a paragraph about high risk individuals for example those with longstanding chronic pancreatitis, with a family history of pancreatic cancer, or those with new onset diabetes mellitus. Otherwise this manuscript provides important and valid information that will be off considerable interest too many clinicians and researchers in this field.

Author Response

Comments and Suggestions for Authors by Reviewer 1

The present manuscript by Shingo Kato and Kazufumi Honda is an updated review of potential biomarkers and imaging modalities for screening and early detection of pancreatic cancer. The manuscript is a well written and comprehensive review of the topic. Analysed aspects include CA19-9, exosomes and microRNA, circulating tumor DNA, circulating tumor cells, multi-analyte blood test, apolipoprotein A2 isoforms, and MRI techniques.

The most important papers are cited and discussed. Somewhat missing is a paragraph about high risk individuals for example those with longstanding chronic pancreatitis, with a family history of pancreatic cancer, or those with new onset diabetes mellitus. Otherwise this manuscript provides important and valid information that will be off considerable interest too many clinicians and researchers in this field.

Response to Reviewer 1

Thank you very much for your positive comments. We added a paragraph describing high-risk individuals, including those with longstanding chronic pancreatitis, with a family history of pancreatic cancer, or those with new-onset diabetes mellitus.

Reviewer 2 Report

First of all, this reviewer would like to congratulate the authors on the effort of writing this interesting review regarding early detection of PDAC.

Although the manuscript is well written there are some concerns that should be addressed:

Overall there seems to be a confusion between technology and biology across the paper regarding subtitles. 11 vs 12 vs 13...A better organisation of the ideas would help to avoid this problem.

Title: the title seems too long. This reviewer would suggest reducing the number of words and make it simple “Non-invasive early detection of pancreatic cancer” or “Use of biomarkers and imaging for early detection of pancreatic cancer”.

Abstract: In general, ok but it is somehow confusing the “invasive and expensive…not feasible”. The authors should specify to which methodology are they referring to or just note that patient selection is the key to make it feasible. As of today, Pathological diagnosis is needed and in early PDAC, Endoscopic ultrasound is needed and expensive.

The organization of the text is too lineal. There are 15 different subtitles, and they should somehow be reduced (eg. 7-8-9-10-11 to 7-7.1-7.2-7.3…). Authors should try to organize in a rational way their different paragraphs. There has to be a common line that is able to grouped them (Blood vs imaging; inside blood it could be protein or DNA/RNA and then insert their items inside them (lipoproteins A2; ctDNA; circulating cells; exosomes or microRNA…)

Point 1 and 2 could be reduced. I think the point of the relevance of PDAC regarding time of diagnosis, OS and incidence in both Japan and the world can be describe in shorter and clearer lines. In addition, in some parts there is a lot of detail (lines 40-44) whereas there are numbers 37% 5y survival in localized PDAC that is not clear if they refer to the USA, the world or where exactly.

Point 3: Not clear for the reader what they want to say. They should reorganize and include panel testing with NCCN and MSKCC (and others) data to explain the strategy. It seems that the last paragraph (lines 99-107) may become the first and then elaborate from there. Maybe comment the BRCA-Olaparib situation that can change the actual management and increase genetic testing along with NCCN guidelines.

Point 4-6 should be 4, 4.1 and 4.2. For PanIN there is some data using autopsy series about the % of patients that do have PanIn when they die. It could be included as a reference for the importance of finding PanINs and how this could also help. It seems that only IPMNs are useful although it is clear that diabetes mellitus can be a selection of high risk patients as it can be chronic pancreatitis. Data regarding different approach like Immunovia (Linda D. Mellby, Andreas P. Nyberg, Julia S. Johansen, Christer Wingren, Børge G. Nordestgaard, Stig E. Bojesen, Breeana L. Mitchell, Brett C. Sheppard, Rosalie C. Sears, and Carl A.K. Borrebaeck. J Clin Oncol. 2018 Aug; 36(28):2887-2894) Could also be added.

In point 7, early detection should be a separate part from risk stratification. Is clear that risk stratification is key for the early detection but they are separate issues. Data from MSKCC genetics studies (among others) are needed to be included, as well as the new NCCN guidelines that recommends genetic testing and could inform families of their risk.

From this part on this reviewer is missing data regarding specificity and sensitivity , AUC and some more comparable statistics between the discussed methods/molecules explained. A table to show differences and explain controls used with the different techniques/molecules (compare Ca19.9 vs ctDNA) and type of patients used (healthy volunteers, diabetes, Pancreatitis chronic or acute, stage of the PDAC patients used on each commented paper…)

Point 13 could be switched just to Non-invasive imaging modalities and then divide 13.1 MRI and 13.1.1 3D analysis of fat….

Some other modalities of non-invasive imaging modalities could also be added here as comparison (CT techniques, ultrasound)

Finally, in the future prospects some ideas about radiomics/radiogenomics could be introduced.  

Author Response

Author's Reply to the Review Report (Reviewer 2)

Comments and Suggestions for Authors by Reviewer 2

First of all, this reviewer would like to congratulate the authors on the effort of writing this interesting review regarding early detection of PDAC.

Although the manuscript is well written there are some concerns that should be addressed:

Response to Reviewer 2

According to your suggestions, we revised our manuscript. Specific changes are described below on a point-by-point basis.

Comment #1

Overall there seems to be a confusion between technology and biology across the paper regarding subtitles. 11 vs 12 vs 13...A better organisation of the ideas would help to avoid this problem.

Response to Comment #1

We revised the subtitles to avoid potential confusion for readers. We created 5 major subtitles with subheadings. We agree that the previous subtitles could be confusing. Thank you for your suggestion.

Comment #2

Title: the title seems too long. This reviewer would suggest reducing the number of words and make it simple “Non-invasive early detection of pancreatic cancer” or “Use of biomarkers and imaging for early detection of pancreatic cancer”.

Response to Comment #2

We revised the title as follows: “Use of biomarkers and imaging for early detection of pancreatic cancer”.

Comment #3

Abstract: In general, ok but it is somehow confusing the “invasive and expensive…not feasible”. The authors should specify to which methodology are they referring to or just note that patient selection is the key to make it feasible. As of today, Pathological diagnosis is needed and in early PDAC, Endoscopic ultrasound is needed and expensive.

Response to Comment #3

We agree with your point. We rephrased the text in question as follows: “However, because of the low incidence of pancreatic cancer in the general population, the stratification of subjects who need to undergo further examinations by invasive and expensive modalities is important.”

Comment #4

The organization of the text is too lineal. There are 15 different subtitles, and they should somehow be reduced (eg. 7-8-9-10-11 to 7-7.1-7.2-7.3…). Authors should try to organize in a rational way their different paragraphs. There has to be a common line that is able to grouped them (Blood vs imaging; inside blood it could be protein or DNA/RNA and then insert their items inside them (lipoproteins A2; ctDNA; circulating cells; exosomes or microRNA…)

Response to Comment #4

We revised the subtitles to reduce the number. Also, some of the text was grouped. Thank you for your suggestion.

Comment #5

Point 1 and 2 could be reduced. I think the point of the relevance of PDAC regarding time of diagnosis, OS and incidence in both Japan and the world can be describe in shorter and clearer lines. In addition, in some parts there is a lot of detail (lines 40-44) whereas there are numbers 37% 5y survival in localized PDAC that is not clear if they refer to the USA, the world or where exactly.

Response to Comment #5

We agree that these parts of the manuscript are redundant. We focused on the US and Japan only in the revised version and omitted the text in lines 40-44 of the original version.

Comment #6

Point 3: Not clear for the reader what they want to say. They should reorganize and include panel testing with NCCN and MSKCC (and others) data to explain the strategy. It seems that the last paragraph (lines 99-107) may become the first and then elaborate from there. Maybe comment the BRCA-Olaparib situation that can change the actual management and increase genetic testing along with NCCN guidelines.

Response to Comment #6

To make the text in question clearer, we revised the subtitle as follows: “Strategy for detecting low-incidence cancers”. We added statements regarding panel testing with NCCN and MSKCC and the BRCA-Olaparib situation in section 1-3.

Comment #7

Point 4-6 should be 4, 4.1 and 4.2. For PanIN there is some data using autopsy series about the % of patients that do have PanIn when they die. It could be included as a reference for the importance of finding PanINs and how this could also help. It seems that only IPMNs are useful although it is clear that diabetes mellitus can be a selection of high risk patients as it can be chronic pancreatitis.

Response to Comment #7

We introduced the results of autopsy studies (refs. 37-42). We agree that autopsy studies are important when discussing the prevalence of PanINs.

Comment #8

Data regarding different approach like Immunovia (Linda D. Mellby, Andreas P. Nyberg, Julia S. Johansen, Christer Wingren, Børge G. Nordestgaard, Stig E. Bojesen, Breeana L. Mitchell, Brett C. Sheppard, Rosalie C. Sears, and Carl A.K. Borrebaeck. J Clin Oncol. 2018 Aug; 36(28):2887-2894) Could also be added.

Response to Comment #8

We added data regarding Immunovia in section 3-4, “Multi-analyte blood test”.

Comment #9

In point 7, early detection should be a separate part from risk stratification. Is clear that risk stratification is key for the early detection but they are separate issues. Data from MSKCC genetics studies (among others) are needed to be included, as well as the new NCCN guidelines that recommends genetic testing and could inform families of their risk.

Response to Comment #9

We added statements regarding panel testing with NCCN and MSKCC in section 1-3.

Comment #10

From this part on this reviewer is missing data regarding specificity and sensitivity , AUC and some more comparable statistics between the discussed methods/molecules explained.

Response to Comment #10

We added data regarding specificity and sensitivity and all available data regarding AUC values.

Comment #11

A table to show differences and explain controls used with the different techniques/molecules (compare Ca19.9 vs ctDNA) and type of patients used (healthy volunteers, diabetes, Pancreatitis chronic or acute, stage of the PDAC patients used on each commented paper…)

Response to Comment #11

We created new Tables 1 and 2. These tables will help readers compare the different techniques/molecules.

Comment #12

Point 13 could be switched just to Non-invasive imaging modalities and then divide 13.1 MRI and 13.1.1 3D analysis of fat….

Response to Comment #12

We revised the subtitles in this part of the manuscript and grouped sections.

Comment #13

Some other modalities of non-invasive imaging modalities could also be added here as comparison (CT techniques, ultrasound)

Response to Comment #13

We added information regarding CT and transabdominal ultrasound in this section.

Comment #14

Finally, in the future prospects some ideas about radiomics/radiogenomics could be introduced.

Response to Comment #14

We added an introduction to radiomics/radiogenomics in the “Future prospects” section.

Thank you very much for your comments.

Reviewer 3 Report

Overall, the review is well written.

Figure 1 is not helpful for the topic of the review and should be left out.

Please prepare instead of Figure 1 a figure or table in the end of the review comparing the different biomarkers and early detection methods, showing their advantages, disadvantages...

Author Response

Author's Reply to the Review Report (Reviewer 3)

Comments and Suggestions for Authors by Reviewer 3

Overall, the review is well written.

Figure 1 is not helpful for the topic of the review and should be left out.

Please prepare instead of Figure 1 a figure or table in the end of the review comparing the different biomarkers and early detection methods, showing their advantages, disadvantages...

Response to Reviewer 3

Thank you very much for your positive comments. In place of the original Figure 1, we prepared a new table comparing the different biomarkers and early detection methods, indicating their advantages and disadvantages.

Round 2

Reviewer 2 Report

Good job. Please review for typos like:

Line 57 ...For example, the although the